# FACTOR: Fairness-Aligned Conformal Transport for Multivariate Mixed Outcomes

## Abstract

In high-stakes domains, decisions often hinge on jointly predicting multiple, correlated outcomes of mixed type (continuous, ordinal, categorical). Existing multivariate conformal methods impose restrictive geometric assumptions, perform poorly with mixed outcomes, or lack subgroup-conditional guarantees, leading to inflated prediction regions and uneven coverage. We propose FACTOR (*Fairness-Aligned Conformal Transport for Optimal Regions*), a framework for constructing compact and equitable prediction regions. FACTOR learns an optimal-transport map in a latent space via normalizing flows with input-convex neural networks, providing a principled multivariate ranking without shape constraints. To enforce fairness, we synchronize latent-space ranks across subgroups, yielding distribution-free marginal coverage and a finite-sample $O(1/N)$ bound on subgroup calibration error. A sliding-window cutoff procedure then minimizes prediction region volume while preserving validity. Empirically, on synthetic and six real-world benchmarks, FACTOR consistently achieves target coverage with reduced region volume and subgroup disparities (measured by KS distance) relative to state-of-the-art baselines under competitive runtime. The method also produces interpretable visualizations and conditional summaries, making FACTOR a practical tool for uncertainty quantification in multivariate, mixed-outcome settings.

## 1 Introduction

**High-stakes, multivariate prediction.** Modern predictive algorithms are increasingly evaluated not on the basis of a single outcome, but by their ability to quantify joint uncertainty across multiple, often correlated outcomes. Consider a few examples. In medicine, regulatory bodies such as the FDA and CDC often make decisions after considering multiple outcomes, including continuous measures of drug efficacy, ordinal toxicity grades, and categorical indicators of safety or adverse events (Meissner, 2022; Dowell, 2022). In education, college admissions committees take into consideration applicants' test scores, ordinal course grades, and other holistic categorical factors (Bastedo et al., 2018; Arcidiacono et al., 2022; Chetty et al., 2023). In economics, central banks set interest rates by jointly monitoring continuous measures like GDP growth and inflation, binary indicators such as recession status, and ordinal measures like credit ratings (Angrist et al., 2018; McAlinn et al., 2020). In all of these settings, decisions are made on the basis of multiple outcomes that are inherently multivariate, span mixed types (continuous, ordinal, categorical), and may be correlated in complex, nonlinear ways. Constructing prediction regions that capture this joint uncertainty in a valid, efficient, and interpretable way is challenging.

**Univariate conformal inference.** Conformal prediction offers a general model-agnostic approach to uncertainty quantification in supervised learning (Vovk et al., 2005; Angelopoulos et al., 2024). Given observations $(X_i, Y_i)_{i=1}^n$ and a new test point $X_{n+1}$, the method constructs a prediction set $C(X_{n+1}) \subseteq \mathcal{Y}$ that satisfies

$$P\big(Y_{n+1} \in C(X_{n+1})\big) \geq 1 - \alpha,$$

for any user-selected miscoverage rate $\alpha \in (0, 1)$. An important strength is that this finite-sample marginal coverage guarantee does not depend on the correctness of the predictive model. Note that the resulting output is a prediction interval for regression problems and a prediction set of labels for classification tasks.

**Challenges in the multivariate setting.** Extending conformal prediction from univariate to multivariate outcomes is not trivial. Multivariate data do not have a direct analogue of ranks or quantiles, which are natural in one dimension and make conformity scores easy to define. Mixed outcome types add another challenge in that the prediction region must respect both continuous and discrete components, which makes simple rectangular or ellipsoidal regions inefficient, and renders copula-based methods unreliable because their smoothness assumptions are violated by construction. These geometric and modeling assumptions fail to reflect the structure of the data and are not able to produce efficient, informative prediction regions. Conversely, methods that focus solely on minimizing prediction region volume are not guaranteed to maintain conditional coverage and can lead to miscoverage for certain subgroups. Finally, many computational approaches rely on nonconvex optimization, which can give unstable solutions and limit scalability.

**Related work.**

SHAPE-CONSTRAINED METHODS. The earliest extensions of conformal prediction for univariate regression to multivariate regression consisted of constructing Cartesian products of marginal prediction intervals, producing hyperrectangles that did not account for correlations among outcome variables and were overly conservative (Neeven & Smirnov, 2018). Ellipsoidal prediction intervals were proposed to incorporate covariance information and produce smaller prediction sets under certain conditions, but they were restricted to convex geometric shapes assuming an underlying elliptical structure and were unable to capture more flexible distributions (Johnstone & Cox, 2021; Messoudi et al., 2022).

COPULA-BASED METHODS. To avoid fixed geometric assumptions, simple parametric copulas have been shown to work for certain datasets (Messoudi et al., 2021). Vine copulas have been proposed to avoid strong parametric assumptions and to directly model dependencies in the outcome distribution (Park et al., 2024), but loss of coverage can occur when the estimated copula of the conformal scores deviates from the true copula, and finite-sample validity cannot be guaranteed (Dheur et al., 2025).

VOLUME-MINIMIZING AND HIGH-DENSITY REGION METHODS. Seeking to minimize volume, Tumu et al. (2024) restrict prediction regions to convex shapes, using heuristic clustering algorithms to adaptively partition the data and maintain coverage. More flexible strategies have been proposed that optimize prediction regions over arbitrary norms, thereby removing restrictive convexity constraints while still achieving exact finite-sample coverage (Braun et al., 2025). However, reliance on first-order optimization techniques introduces the risk of convergence to poor local minima, and aggressive volume reduction can compromise conditional coverage across subgroups. A related line of work focuses on high-density regions (HDR), which define prediction regions as estimated density level sets (Izbicki et al., 2022; Dheur et al., 2024; Jonkers et al., 2025). These methods can produce relatively efficient regions, but they rely on accurate density estimation and do not provide exact finite-sample coverage in the presence of mixed outcomes.

LATENT-SPACE QUANTILE METHODS. A recent approach is to first map the conditional distribution of the response into a latent space where the level sets of the density are convex, and then transform these sets back into the original space. This can be achieved using a deep generative model that learns a latent representation of the response with an approximately unimodal distribution, e.g., using directional quantile regression and conditional variational autoencoders (CVAE). The spherically transformed directional quantile regression (ST-DQR) method of Feldman et al. (2023) can produce smaller prediction regions, but a potential limitation is that its performance depends heavily on the quality of the CVAE, which can be improved by incorporating more modern generative models such as normalizing flows (Kobyzev et al., 2020). Related probabilistic generative approaches fit a conditional generative model for the outcome and construct prediction sets by retaining sampled responses with the largest estimated densities (Wang et al., 2023).

OPTIMAL TRANSPORT METHODS. Our work is most similar to recent optimal transport (OT) methods, which seek to define a meaningful ordering in multidimensional spaces (Chernozhukov et al., 2017; Hallin et al., 2021; 2023). Thurin et al. (2025) and Klein et al. (2025) extended conformal inference techniques to multivariate conformal score functions by transporting the response dis-

tribution to a uniform reference measure using an OT map. Computationally, Klein et al. (2025) uses general entropic maps and establishes finite-sample coverage guarantees. Although these methods introduce OT-based scores, they focus exclusively on marginal coverage, do not incorporate mechanisms for subgroup-conditional guarantees, are not tailored to mixed discrete–continuous outcomes, and do not optimize region volume in the outcome space.

A related line of work learns a transformation into a simple reference distribution using normalizing flows. CONTRA (Fang et al., 2025) uses a real-valued non-volume preserving (RealNVP) bijective flow (Dinh et al., 2017) to push the response toward a Gaussian reference distribution and defines the conformity score as the Euclidean distance to the origin in the transformed space. A variant of CONTRA, called ResCONTRA, attempts to improve predictive performance by training a second normalizing flow on the residuals. However, this approach is less computationally efficient because it requires fitting two complex models on the same dataset. In contrast, our method trains a single flow and additionally introduces a functional synchronization step to guarantee subgroup-conditional coverage for fairness, a feature that is not addressed in CONTRA. The method we develop also transforms the response into a simpler latent space using the gradient of an Input Convex Neural Network (ICNN) to approximate the OT map, which has been shown to provide universal approximation guarantees for convex functions (Chen et al., 2019) and their gradients (Huang et al., 2020).

**Empirical preview.** Table 1 provides a preview of our contributions. We compare three representative baselines—marginal conformal prediction (MCP), a highest-density region method (HDR), and latent conformal prediction (L-CP)—to our proposed method, FACTOR. The table highlights three key takeaways: (i) FACTOR achieves prediction regions with the average size on par with methods designed specifically to minimize volume, e.g., HDR; (ii) FACTOR markedly improves fairness, as shown by reduced subgroup disparities measured by Kolmogorov–Smirnov distance; and (iii) these gains come at computational cost comparable to density/region-estimation baselines such as HDR. This preview illustrates the advantages of combining conformal inference, optimal transport, and fairness calibration in a unified framework.

Table 1: Comparison of methods: Log of average region size (LogAvgSize), Kolmogorov–Smirnov (KS) distance, Empirical coverage, and Elapsed time (Time) in seconds for for two-dimensional outcomes: Price (continuous) and Grade (discrete), across three levels of Floor (subgroup) in the dataset "house" from Feldman et al. (2023)

| Method | Log(AvgSize) | KS distance | Coverage | Time (s) |
|---|---|---|---|---|
| MCP | 16.56 | 0.06 | (0.97, 0.91, 0.94) | 75.91 |
| HDR | **15.85** | 0.05 | (0.99, 0.96, 0.94) | 70.22 |
| L-CP | 15.86 | 0.03 | (0.96, 0.93, 0.96) | **6.55** |
| FACTOR (ours) | 15.90 | **0.00** | **(0.94, 0.94, 0.94)** | 70.79 |

**Our contributions.** This work introduces FACTOR (*Fairness-Aligned Conformal Transport for Optimal Regions*), a framework that addresses the limitations of prior approaches. FACTOR learns an OT map that pushes mixed-type multivariate outcomes into a simple latent space, estimated using normalizing flows with input-convex neural networks (ICNNs). The induced transport distance defines a scalar multivariate rank that accommodates both continuous and discrete outcomes. Although robustness of learned OT maps is an active research area, several useful approximation results exist. Prior work shows that ICNNs can approximate any convex function Chen et al. (2019) and that their gradients can approximate monotone multivariate maps Huang et al. (2020). These results imply that, under mild regularity assumptions, an ICNN-based flow can approximate the optimal OT map universally with small training error. To ensure subgroup calibration, FACTOR introduces a synchronization step that aligns the distribution of ranks across subgroups, yielding explicit finite-sample bounds on calibration error. Finally, a sliding-window cutoff optimization procedure minimizes prediction region volume subject to validity.

FACTOR satisfies distribution-free marginal coverage for arbitrary mixed-type outcomes; further, we provide explicit finite-sample subgroup calibration guarantees, showing that groupwise coverage disparities decay at rate $O(1/N)$. We develop a scalable implementation based on OT maps and

ICNN-powered normalizing flows, together with the synchronization procedure and efficient cutoff search. Empirically, FACTOR consistently achieves target coverage, reduces prediction region volume relative to state-of-the-art methods, and lowers subgroup disparities, while maintaining runtime comparable to density/region-estimation baselines. These findings hold in both controlled synthetic experiments and evaluations on six publicly available benchmark datasets spanning different disciplines.

## 2 METHOD

**Problem setup and goal.** Our goal is to construct valid, compact, and equitable prediction regions for multivariate, mixed-type outcomes. Let $X \in \mathbb{R}^d$ denote covariates, $S \in \{1, \dots, K\}$ a protected subgroup label, referring to the value of the sensitive attribute (e.g., race, gender, socioeconomic group), and $Y \in \mathcal{Y}$ the outcomes of interest, where $\mathcal{Y}$ may contain both continuous and discrete components.

In the univariate case, conformal prediction relies on ranks or quantiles of conformal scores to form intervals or label sets. However, once $p > 1$, there is no canonical analogue of a rank or quantile. This lack of a natural ordering makes it unclear how to calibrate prediction regions in a way that is both valid and efficient. Naive fixes such as using Euclidean distance to rank outcomes quickly break down: skewed or correlated distributions distort distances, leading to overly large or misshapen regions (Klein et al., 2025). The central challenge, then, is to define a multivariate ranking that respects the joint structure of mixed outcomes and supports fairness across subgroups. Our solution builds on optimal transport to construct such ranks, which we detail next.

### 2.1 OPTIMAL TRANSPORT FOR MULTIVARIATE PREDICTION

To address the absence of a canonical multivariate rank, we adopt an optimal transport (OT) perspective. Let $\mathbb{P}_{Y|X,S}$ denote the conditional outcome distribution. The OT map $q^*(Y, X, S)$ pushes $\mathbb{P}_{Y|X,S}$ forward to the uniform distribution $\mathbb{U}^p$ on the unit ball $B(0, 1)$ by minimizing average transportation cost:

$$q^*(Y, X, S) = \arg \min_{q : q(Y,X,S) \sim \mathbb{U}^p} \int_\Omega \|y - q(y, x, s)\|^2 d\mathbb{P}_{Y,X,S}, \tag{1}$$

where the integral is with respect to the joint law of $(Y, X, S)$. By Brenier's theorem (Brenier, 1991), such a map exists when $\mathbb{P}_{Y|X,S}$ admits a density.

Define the transported distance $u^*(Y, X, S) = \|q^*(Y, X, S)\|$ as the raw conformal score. For a new point $(X, S)$, the conformal prediction set is

$$C_\alpha^0(X, S) = \{y \in \mathcal{Y} : u^*(y, X, S) \le r_\alpha\}, \tag{2}$$

where $r_\alpha$ is the $(1 - \alpha)$ empirical quantile of $\{u^*(Y_i, X_i, S_i)\}$.

**Theorem 1** (Coverage Guarantee)**.** *For any OT map $q^*(\cdot)$, the prediction set $C_\alpha^0(X, S)$ satisfies*

$$P\{Y \in C_\alpha^0(X, S)\} \ge 1 - \alpha,$$

*with no distributional assumptions beyond exchangeability.*

**Remark 1.** *In the univariate case, $u^*(\cdot)$ reduces to the CDF $F(\cdot)$, so the OT-based rank is a direct multivariate generalization of quantiles.*

### 2.2 FUNCTIONAL SYNCHRONIZATION FOR GROUP COVERAGE

**From marginal to subgroup coverage.** The sets $C_\alpha^0(X, S)$ guarantee marginal coverage but not subgroup-conditional coverage. In practice, prediction sets may be systematically miscalibrated across protected groups. To address this, we project the raw conformal score $u^*(Y, X, S)$ onto the class of fair functions

$$\mathcal{G} = \Big\{ v : \sup_t \big| P(v(Y, X, S) \le t \mid S = s) - P(v(Y, X, S) \le t \mid S = s') \big| = 0, \ \forall s \ne s' \Big\},$$

which enforces demographic parity across subgroups (Gouic et al., 2020). Other fairness notions can be incorporated by modifying $\mathcal{G}$.

Following Chzhen et al. (2020), we define the fair conformal score $v^*(Y, X, S)$ as the $L^2$-projection of $u^*(Y, X, S)$ onto $\mathcal{G}$:

$$v^*(Y, X, S) = \arg\min_{v \in \mathcal{G}} \mathbb{E}\big[|u^*(Y, X, S) - v(Y, X, S)|^2\big] \tag{3}$$

$$= \arg\min_{v \in \mathcal{G}} \sum_{s=1}^{K} p_s \int_{\Omega} |u^*(Y, X, s) - v(Y, X, s)|^2 d\mathbb{P}, \tag{4}$$

where $p_s = P(S = s)$.

**Theorem 2** (Fair OT Map). *The minimizer has the closed form*

$$v^*(Y, X, S) = \left( \sum_{s'=1}^{K} p_{s'} Q_{u^*|s'} \right) \circ F_{u^*|s}\big(u^*(Y, X, S)\big),$$

*where $F_{u^*|s}$ and $Q_{u^*|s'}$ are the subgroup-specific CDF and quantile functions of $u^*$.*

**Remark 2.** *This synchronization step aligns the distribution of ranks across groups while preserving their ordering within each group. Importantly, the OT map need not be trained separately for each subgroup. Synchronization is performed only once post hoc on the raw conformal scores.*

## 2.3 CUTOFF OPTIMIZATION FOR VOLUME OPTIMALITY

**Level-set prediction regions.** Given the fair conformal score $v^*(Y, X, S)$, we define prediction sets as level sets:

$$C_\alpha(X, S) = \{y \in \mathcal{Y} : r_{\alpha,1} \leq v^*(y, X, S) \leq r_{\alpha,2}\}. \tag{5}$$

In the population, $v^*(Y, X, S)$ admits a simple limiting law. If $q^*$ is the Brenier map pushing $\mathbb{P}_{Y|X,S}$ to $\mathbb{U}^p$, then $q^*(Y, X, S)$ is uniform on the $p$-ball, and its radial component $u^*(Y, X, S) = \|q^*(Y, X, S)\|$ satisfies $u^*(Y, X, S)^p \sim \mathrm{Unif}[0, 1]$. Thus, the quantiles of $\mathrm{Unif}[0, 1]$ serve as natural cutoffs for $v^*(Y, X, S)$ as the $v^* \to u^*$.

**From one-sided to shortest interval.** A simple baseline is the one-sided rule $[0, \widehat{Q}_{1-\alpha}]$, where $\widehat{Q}_{1-\alpha}$ is the $(1 - \alpha)$ empirical quantile of $\{v^*(Y_i, X_i, S_i)\}$. While always valid, this choice can be inefficient when finite-sample noise or discreteness produces irregularities in the empirical rank distribution. To reduce volume, we instead solve

$$C_\alpha^{\mathrm{opt}}(X, S) = \arg\min_{C_\alpha(\cdot)} \left\{ |C_\alpha| : P\big(Y \in C_\alpha(X, S)\big) \geq 1 - \alpha \right\}$$

$$= \arg\min_{r_{\alpha,1}, r_{\alpha,2}} \left\{ |C_\alpha| : P(r_{\alpha,1} \leq v^*(Y, X, S) \leq r_{\alpha,2}) \geq 1 - \alpha \right\}. \tag{6}$$

This program seeks the smallest prediction regions in the outcome space on average by optimizing the cutoff values in the rank space with guaranteed coverage.

**Remark 3.** *For the uniform distribution, every interval of width $1 - \alpha$ covers the same mass, so $[0, 1 - \alpha]$ is a valid option. However, such an interval might not be optimal, as it can contain points in the outcome space with very small probabilities. Allowing both $r_{\alpha,1}$ and $r_{\alpha,2}$ to vary retains only the $(1 - \alpha) \times 100\%$ of samples with the highest density in the outcome space, ensuring that the prediction region is concentrated on high-density points and leads to smaller, more efficient prediction regions.*

## 3 ALGORITHMIC IMPLEMENTATION

The theoretical framework above relies on the population OT map $q^*(\cdot)$, which is unknown in practice. We approximate it from finite samples using *normalizing flows* (Kan et al., 2022). A normalizing flow learns an invertible transformation $q$ such that

$$p(Y, X, S) \approx p_U\big(q(y, x, s)\big) \, \det\left( \frac{\partial q(y, x, s)}{\partial y} \right) \equiv p_q(y, x, s),$$

mapping an outcome $Y$ with density $\mathbb{P}_{Y|X,S}$ to a uniform latent variable $U$.

**Why normalizing flows.** Normalizing flows are particularly suited to our setting: their invertibility allows us to both push outcomes forward into a uniform latent space and pull calibrated ranks back into outcome space for prediction. The tractable Jacobian makes likelihood-based training feasible via the KL divergence, unlike GANs or diffusion models, which lack explicit densities. Compared to VAEs, flows avoid approximate inference, providing exact likelihoods. Moreover, when combined with ICNNs, flows can approximate convex transport maps that enforce monotonicity while flexibly modeling nonlinear dependencies with theoretical guarantees (Chen et al., 2019; Huang et al., 2020).

**Training the transport map.** We train $q_\theta$ by minimizing the KL divergence between $p_Y$ and $p_q$:

$$\mathrm{KL}(p_{Y,X,S}\|p_q) = \mathbb{E}_{y\sim p_Y}\Big[\log \tfrac{p(y,x,s)}{p_q(y,x,s)}\Big] = \mathbb{E}_{y\sim p_Y}\{\log p(y,x,s) - \log p_q(y,x,s)\},$$

where the first term does not depend on the OT map $q(\cdot)$, and the empirical version for the second term is

$$\frac{1}{n}\sum_{i=1}^{n}\Big[-\log p_U\big(q(y_i,x_i,s_i)\big) - \log\det\Big(\tfrac{\partial q(y_i,x_i,s_i)}{\partial y}\Big)\Big]. \tag{7}$$

The connection between this KL objective and quadratic optimal transport is established in (Brenier, 1991) via the Knott–Smith criterion. To enforce monotonicity, we parameterize $q_\theta(Y,X,S) = \nabla_y G_\theta(Y,X,S)$ as the gradient of an ICNN, ensuring that $q_\theta$ approximates valid transport maps.

**Split conformal strategy.** Following the split conformal framework, we partition the sample into training $\mathcal{D}_{\mathrm{tr}}$ and calibration $\mathcal{D}_{\mathrm{cal}}$. On $\mathcal{D}_{\mathrm{tr}}$, the OT map $\widehat{q}_\theta$ is fit by minimizing

$$\widehat{q}_\theta = \arg\min_\theta \sum_{i\in\mathcal{D}_{\mathrm{tr}}}\Big[-\log\det\big(\tfrac{\partial q_\theta(y_i,x_i,s_i)}{\partial y}\big)\Big],$$

where the uniform density $p_U$ drops out because it is constant on its support (the unit $p$-ball) and thus independent of $\theta$.

**Fair conformal scores.** On the calibration set, we compute the raw conformal scores by

$$\widehat{u}(Y_i,X_i,S_i) = \big\|\widehat{q}_\theta(Y_i,X_i,S_i)\big\|, \quad i\in\mathcal{D}_{\mathrm{cal}},$$

and synchronize them across subgroups to obtain its fair version for group-conditional coverage

$$\widehat{v}(Y,X,S) = \Big(\sum_{s'=1}^{K} p_{s'}\,\widehat{Q}_{\widehat{u}|s'}\Big)\circ\widehat{F}_{\widehat{u}|S}\big(\widehat{u}(Y,X,S)\big),$$

where $\widehat{F}_{\widehat{u}|s}$ and $\widehat{Q}_{\widehat{u}|s}$ are the empirical CDF and quantile functions of $\{\widehat{u}(Y_i,X_i,S_i): S_i = s\}$.

**Prediction sets.** Conformal prediction is then applied to $\widehat{v}$. Since $q_\theta$ is invertible, prediction sets can be pulled back into outcome space:

$$\widehat{C}_\alpha(X,S) = \Big\{y\in\mathcal{Y}: \widehat{r}_{\alpha 1}\le\widehat{v}(y,X,S)\le\widehat{r}_{\alpha 2}\Big\},$$

with cutoffs $\widehat{r}_{\alpha 1},\widehat{r}_{\alpha 2}$ selected by minimizing the expected region size $|C_\alpha|$ with the idea of importance sampling:

$$|C_\alpha| = \int \mathbf{1}(Y\in C_\alpha(X,S))d\mathbb{P}_{Y,X,S} = \mathbb{E}\left\{\frac{\mathbf{1}(Y\in C_\alpha(X,S))}{p(Y,X,S)}\right\}$$

$$\approx \frac{1}{|\mathcal{D}_{\mathrm{cal}}|}\sum_{i\in\mathcal{D}_{\mathrm{cal}}}\frac{\mathbf{1}(r_{\alpha 1}\le\widehat{v}(Y_i,X_i,S_i)\le r_{\alpha 2})}{p_{\widehat{q}_\theta}(Y_i,X_i,S_i)}.$$

The intuition is that the interval achieving the desired $\alpha$-coverage should contain as few outcomes as possible, where each outcome is weighted inversely by its probability, thereby minimizing the region size in the outcome space.

---

**Algorithm 1** Fairness-Aligned Conformal Transport for Optimal Regions (FACTOR)

---

**Input:** data $\{(X_i, S_i, Y_i)\}_{i=1}^N$, miscoverage level $\alpha \in (0, 1)$
Split data into training $\mathcal{I}_1$, testing $\mathcal{I}_2$, and calibration $\mathcal{I}_3$.
**Step 1: Train OT map.** Train $\widehat{q}_\theta$ on $\mathcal{D}_{\mathrm{tr}} = \mathcal{I}_1 \cup \mathcal{I}_2$.
**Step 2: Transported distances.** Compute $\widehat{u}(Y, X, S) = \|\widehat{q}_\theta(Y, X, S)\|$ on $\mathcal{D}_{\mathrm{cal}} = \mathcal{I}_3$.
**Step 3: Synchronization.** Align subgroup distributions to obtain $\widehat{v}(Y, X, S)$.
**Step 4: Cutoff optimization.** Find cutoffs $\widehat{r}_{\alpha1}, \widehat{r}_{\alpha2}$ by minimizing the region size.
**Output:** $\widehat{C}_\alpha(X, S) = \{y \in \mathcal{Y} : \widehat{r}_{\alpha1} \leq \widehat{v}(y, X, S) \leq \widehat{r}_{\alpha2}\}$ for new $(X, S)$.

---

**Algorithm summary.** The complete procedure, FACTOR (*Fairness-Aligned Conformal Transport for Optimal Regions*), is summarized in Algorithm 1.

## 4 THEORETICAL PROPERTIES

**Theorem 3** (Empirical group-conditional coverage). *Let $N_s = \sum_{i \in \mathcal{D}_{\mathrm{cal}}} \mathbf{1}(S_i = s)$ be the calibration sample size for subgroup $s$. Then for any $s \neq s' \in \{1, \ldots, K\}$, the synchronized transported distance $\widehat{v}(Y, S)$ satisfies*

$$\sup_t \left| P(\widehat{v}(Y, X, S) \leq t \mid S = s) - P(\widehat{v}(Y, X, S) \leq t \mid S = s') \right| \leq \frac{1}{\min\{N_s, N_{s'}\} + 1}.$$

**Remark 4.** *Theorem 3 provides a finite-sample fairness guarantee: subgroup calibration error is bounded at order $O(1/\min\{N_s, N_{s'}\})$. Two implications follow. First, FACTOR achieves near-exact subgroup calibration even with moderate calibration set size. Second, the bound highlights the role of subgroup balance: guarantees are strongest when calibration counts $N_s$ are similar across groups, motivating stratified sampling or reweighting to avoid imbalances in practice.*

**Corollary 1** (Asymptotic subgroup validity). *If $\min_{s \in \{1, \ldots, K\}} N_s \to \infty$, then*

$$\sup_t \left| P(\widehat{v}(Y, X, S) \leq t \mid S = s) - P(\widehat{v}(Y, X, S) \leq t \mid S = s') \right| \to 0, \quad \text{for all } s \neq s'.$$

*In particular, if subgroup proportions in the calibration set satisfy $\inf_s P(S = s) > 0$, then $|\mathcal{D}_{\mathrm{cal}}| \to \infty$ implies $\min_s N_s \to \infty$, and FACTOR achieves subgroup-conditional calibration asymptotically.*

**Remark 5.** *The conclusion holds under the same exchangeability conditions as conformal prediction and consistency of the learned transport $\widehat{q}_\theta$ (so that the empirical rank law of $\widehat{v}$ converges to that of $v^*$). The finite-sample bound in Theorem 3 already yields an $O(1/\min\{N_s, N_{s'}\})$ rate; the corollary follows by letting $\min_s N_s \to \infty$.*

## 5 EXPERIMENTS

**Baseline and metrics.** We evaluate FACTOR on both synthetic data and six real-world benchmark datasets, comparing against three representative conformal methods: (1) marginal conformal prediction (MCP), (2) the conditional highest predictive density method (HDR), (3) latent conformal prediction (L-CP), (4) transformed directional quantile regression (ST-DQR) in Feldman et al. (2023), (5) probabilistic conformal prediction (PCP), and (6) high-density conformal prediction (HD-CP) in Wang et al. (2023). These baselines span the main approaches in the literature: marginalization, highest-density sets, and latent-space conformalization.

Performance is measured along three dimensions: (i) *average prediction region size*, which captures efficiency, (ii) *Kolmogorov–Smirnov (KS) distance*, which quantifies subgroup fairness by measuring the sup CDF difference of rank distributions across subgroups $s \in \{1, ..., K\}$, and (iii) *average elapsed time*, which reflects computational complexity. Additional measurements and ablation study are provided in the Appendix.

### 5.1 SYNTHETIC EXPERIMENTS

**Setup.** In the synthetic experiments, we vary outcome correlations and subgroup-specific variances to examine how each method performs under controlled conditions. We generate data as

follows:

$$X \sim U[0.5, 1], \quad S \sim \text{Categorical}(\{1, 2, 3\}; 1/3, 1/3, 1/3), \quad Y \mid X, S \sim N(\mathbf{0}_p, XS\Sigma_Y),$$

where $\Sigma_Y = \begin{bmatrix} 1 & \rho \\ \rho & 1 \end{bmatrix}$ with $\rho \in \{0, 0.5, 0.8\}$. This design allows us to study how methods behave as correlations increase and as subgroup-specific variances scale with $S$.

For prediction models we employ the Multivariate Quantile Function Forecaster (MQF$^2$), a normalizing-flow–based model that directly estimates multivariate conditional quantile functions by ICNNs (Kan et al., 2022; Dheur et al., 2025), trained within the split conformal framework (40% training, 20% testing, 40% calibration).

**Results.** Figure 1 summarizes efficiency, fairness, and runtime. As expected, MCP produces the largest prediction regions for strongly correlated outcomes (e.g., $\rho = 0.8$) because it treats outcomes independently. HDR yields tighter prediction regions but is computationally expensive, requiring many samples to approximate highest-density regions. L-CP is faster but does not directly enforce fairness, leading to larger subgroup disparities. By contrast, FACTOR consistently achieves the smallest prediction regions among fairness-preserving methods, maintains low KS distance across subgroups, and runs with reasonable computational cost. The illustrative examples in the middle and bottom panels highlight how FACTOR adapts to correlation $\rho = 0.8$, yielding compact and balanced subgroup coverage.

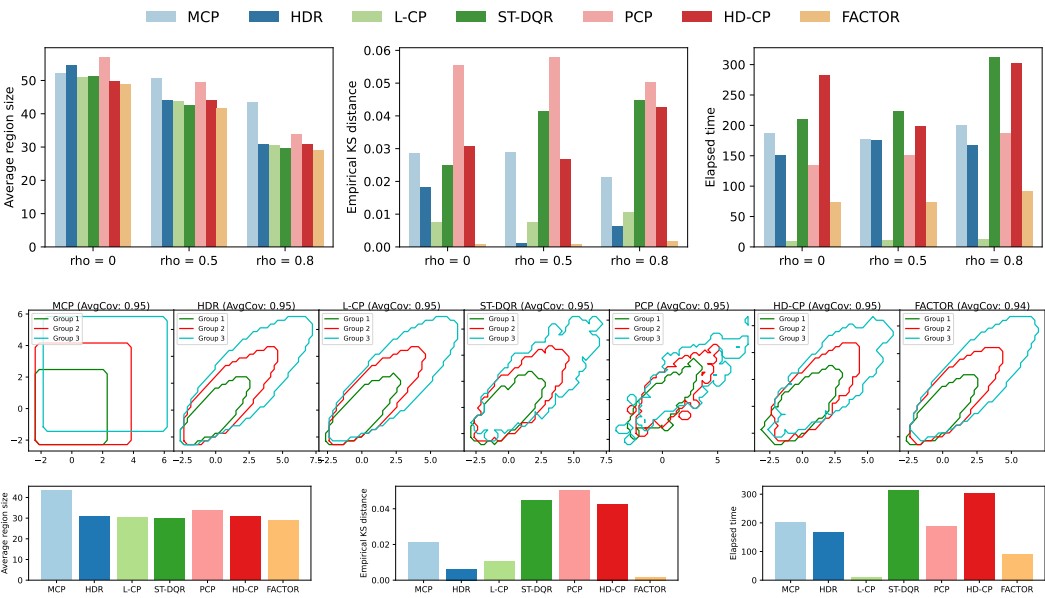

Figure 1: (Top) Average region size, empirical Kolmogorov–Smirnov distance, and elapsed time for multivariate conformal methods on bivariate normal outcomes ($N = 5000$) with correlations $\rho = 0, 0.5, 0.8$. (Middle) Example regions given one new draw $(\bar{X}, \bar{S})$ as the average of the calibration set when $\rho = 0.8$. (Bottom) Average model performance for the calibration set: FACTOR achieves compact prediction regions while maintaining subgroup fairness.

## 5.2 REAL-WORLD DATASETS

**Datasets.** We further benchmark on six publicly available datasets used in prior conformal prediction studies. These datasets span diverse domains, with sample sizes between 768 and 21,613, covariate dimensions $d \in [5, 77]$, outcome dimensions $p$ ranging from 2 to 6, and protected subgroups ranging from 2 to 8. This variety enables us to test scalability, fairness, and efficiency in settings that more closely reflect real applications. Detailed descriptions of each dataset and preprocessing steps are provided in the Appendix.

**Results.** Figure 2 reports the same three metrics as in the synthetic experiments. Across all datasets, FACTOR achieves a favorable balance: prediction regions are consistently smaller than or on par with those of fairness-agnostic baselines while always maintaining subgroup coverage parity (low KS distance). Computational cost remains competitive, with runtime close to L-CP and much faster than HDR for larger datasets (e.g. air and wage). These results confirm that the theoretical guarantees of FACTOR translate into tangible gains across a range of practical tasks.

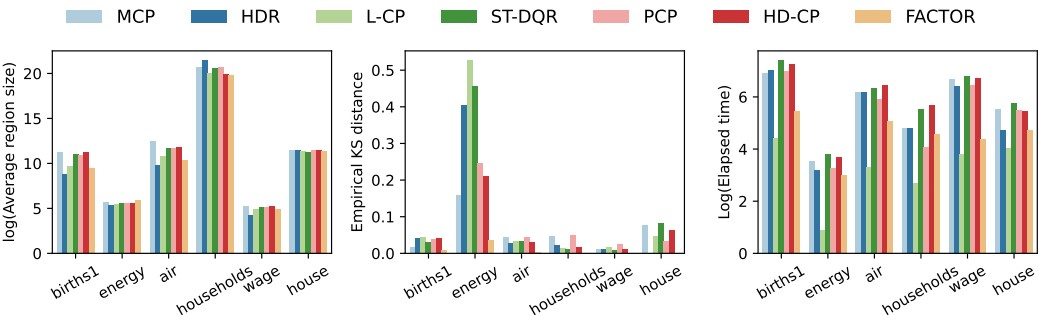

Figure 2: Average prediction region size, subgroup KS distance, and elapsed time for multivariate conformal methods across six real-world datasets. FACTOR yields compact and fair prediction regions while maintaining competitive runtime.

## 6 DISCUSSION

**Summary.** A central challenge in multivariate conformal prediction is to construct prediction regions that are both compact and valid. Following Feldman et al. (2023), this goal is closely linked to estimating the conditional distribution $P(Y \mid X)$ with high fidelity. Early approaches based on marginal quantile regression yield Cartesian-product prediction sets, which are conservative and often inefficient. More recent methods such as directional multivariate quantile regression (Dheur et al., 2025) improve efficiency by estimating boundary quantiles along multiple directions, but they require careful directional choices and can be computationally demanding.

**Contributions.** FACTOR departs from these strategies. By employing flexible base predictors that handle both discrete and continuous outcomes, it maps complex, mixed-type outcome distributions into latent spaces with simpler (often unimodal) structure. Conformalization is then performed in this latent space, eliminating the need for grid searches or directional quantile estimation. The framework also naturally accommodates divergence-based training objectives such as KL divergence, which extend applicability to discrete outcomes.

A distinctive contribution of FACTOR is its synchronization step, which enforces subgroup-conditional calibration. The finite-sample guarantee in Theorem 3 bounds calibration error across groups at rate $O(1/N)$, which implies that subgroup fairness improves rapidly with calibration sample size and converges to exact calibration asymptotically. The explicit dependence on subgroup sample sizes underscores the importance of balanced calibration, motivating stratified designs or reweighting strategies. Empirically, FACTOR achieves markedly lower subgroup disparities than existing methods while maintaining compact prediction regions and competitive runtime.

**Limitations.** However, limitations remain. FACTOR assumes reasonably well-specified base predictors, and systematic robustness analysis under model misspecification is an important direction for future work. While the method scales effectively to moderate-dimensional outcomes, extending it to ultra–high-dimensional response spaces may require additional structure, such as sparsity or low-rank assumptions. Our fairness criterion is based on subgroup-conditional coverage; extending the framework to other notions of fairness, such as continuous sensitive attributes, equalized odds, or calibration in intersectional subgroups, would broaden appeal. Moreover, practical deployment

will require integration with domain knowledge expertise, particularly in medicine and economics, where interpretability and regulatory oversight are critical.

**Future work.**  Several extensions appear promising. In multi-source and federated settings, where data are distributed across sites with heterogeneous distributions, conformal prediction must address distribution shift and privacy constraints. Recent advances in multi-source conformal inference (Liu et al., 2024) and robust learning under distribution shift in clinical AI (Han, 2025) provide a foundation for extending FACTOR to these contexts, with fairness guarantees that remain valid across sources. Another promising direction is leveraging surrogate outcomes to improve efficiency: surrogate-assisted conformal methods (Gao et al., 2025; 2024) show that region size can be reduced without sacrificing validity, which could be especially valuable when primary outcomes are rare or costly to measure. Finally, there is growing research in combining conformal prediction with causal inference, particularly for individualized treatment effects (Lei & Candès, 2021). Extending FACTOR to causal settings is of interest for obtaining valid, fair, and interpretable uncertainty quantification for decision-making in comparative effectiveness research.

## ETHICS STATEMENT

This work complies with the ICLR Code of Ethics. We used only publicly available datasets with appropriate licenses and did not involve human subjects or sensitive personal information. We acknowledge potential risks of misuse (e.g., unfair application, misinterpretation, or unintended deployment beyond the intended research scope) and discuss limitations and safeguards in the paper. All results are reported transparently, and code will be released to support reproducibility.

## REPRODUCIBILITY STATEMENT

All simulation studies and real data analysis were performed using Python version 3.13. All source code and software (Python package) will be made publicly available through the author's GitHub upon acceptance of the paper.

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

# A APPENDIX

## A.1 LLM USAGE STATEMENT

We acknowledge the use of ChatGPT-5.0 exclusively for language polishing and grammatical corrections. No large language models (LLMs) were used for any other aspects of this work. The research ideas, conceptualization, methodology development, and all experiments are entirely original contributions of the authors.

## A.2 PROOF OF THEOREM 2

*Proof.* Let $\mu_s := \mathcal{L}(u^*(Y, X, S) \mid S = s)$ denote the conditional law of $u^*$ in subgroup $s$, and let $p_s = P(S = s)$. Recall that $\mathcal{G}$ is the class of measurable functions $v(Y, X, S)$ whose marginal distributions are identical across subgroups, i.e., $\mathcal{L}(v \mid S = s) = \mu$ for all $s$ and some common $\mu$.

**Step 1 (Projection $\Longleftrightarrow$ barycenter).** For any candidate $v \in \mathcal{G}$ with common marginal $\mu$, the $L^2$ projection objective decomposes by subgroups:

$$\mathbb{E}\Big[\big(u^* - v\big)^2\Big] = \sum_{s=1}^{K} p_s \, \mathbb{E}\Big[\big(u^* - v\big)^2 \,\Big|\, S = s\Big].$$

On the real line, the $L^2$-optimal coupling between $\mu_s$ and $\mu$ is the monotone (quantile) coupling. Hence, for each $s$, the minimizer over all measurable maps with $\mathcal{L}(v \mid S = s) = \mu$ is

$$v_s^\star \;=\; Q_\mu \circ F_{\mu_s}(u^*),$$

and the minimal value of the subgroup term equals $W_2^2(\mu_s, \mu)$, the squared 1-D Wasserstein distance. Therefore

$$\min_{v \in \mathcal{G}} \mathbb{E}\Big[\big(u^* - v\big)^2\Big] \;=\; \min_{\mu} \sum_{s=1}^{K} p_s \, W_2^2(\mu_s, \mu),$$

which is exactly the Wasserstein–2 barycenter problem on $\mathbb{R}$ (Chzhen et al., 2020, Theorem 2.3).

**Step 2 (Closed form via quantiles).** In one dimension, the $W_2$ barycenter has an explicit quantile function:

$$Q_{\mu^\star}(t) \;=\; \sum_{s=1}^{K} p_s \, Q_{\mu_s}(t), \qquad t \in [0, 1],$$

see Chzhen et al. (2020, Lemma A.2). Plugging the barycenter $\mu^\star$ back into the subgroupwise monotone couplings from Step 1 yields the fair projection

$$v^*(Y, X, s) \;=\; Q_{\mu^\star}\big(F_{\mu_s}(u^*(Y, X, s))\big) \;=\; \Big( \sum_{s'=1}^{K} p_{s'} \, Q_{\mu_{s'}} \Big) \circ F_{\mu_s}\big(u^*(Y, X, s)\big),$$

which matches the claimed expression with $F_{u^*|s} = F_{\mu_s}$ and $Q_{u^*|s} = Q_{\mu_s}$. This $v^*$ attains the minimum because (i) for the common marginal $\mu^\star$ it uses the subgroupwise optimal (monotone) couplings, and (ii) $\mu^\star$ minimizes the weighted sum of $W_2^2$ distances over $\mu$. $\qquad\square$

### A.3 PROOF OF THEOREM 3

*Proof.* The proof is adapted from Proposition 4.1, Chzhen et al. (2020). Fix distinct subgroups $s \neq s'$. Let $\mathcal{I}_s = \{i \in \mathcal{D}_{\text{cal}} : S_i = s\}$ and $N_s = |\mathcal{I}_s|$; define analogously $\mathcal{I}_{s'}$, $N_{s'}$. Write

$$\widehat{u}_i = \|\widehat{q}_\theta(Y_i, X_i, S_i)\|, \quad \widehat{F}_s(t) = \frac{1}{N_s} \sum_{i \in \mathcal{I}_s} \mathbf{1}\{\widehat{u}_i \leq t\}$$

for the empirical CDF of transported distances $\widehat{u}_i$ within subgroup $s$. By construction of the synchronization map,

$$\widehat{v}(Y, X, s) = \Big( \sum_{r=1}^{K} p_r \, \widehat{Q}_{\widehat{u}|r} \Big) \circ \widehat{F}_s \big( \widehat{u}(Y, X, s) \big),$$

which is a composition of $\widehat{F}_s$ with a nondecreasing transform that does not depend on $s$ except through its argument. Hence, for any $t$,

$$\{\widehat{v}(Y, X, s) \leq t\} \;\Leftrightarrow\; \{\widehat{F}_s(\widehat{u}(Y, X, s)) \leq \psi(t)\},$$

for some nondecreasing $\psi$ independent of $s$. Therefore,

$$\sup_t \Big| P\big(\widehat{v}(Y, X, s) \leq t \mid S = s\big) - P\big(\widehat{v}(Y, X, s') \leq t \mid S = s'\big) \Big|$$

$$= \sup_t \Big| P\big(\widehat{F}_s(\widehat{u}(Y, X, s)) \leq t \mid S = s\big) - P\big(\widehat{F}_{s'}(\widehat{u}(Y, X, s')) \leq t \mid S = s'\big) \Big|.$$

It thus suffices to bound the Kolmogorov distance between the laws of $\widehat{F}_s(\widehat{u})$ across subgroups.

**Rank-uniformity within a subgroup.** Condition on the calibration multiset $\{\widehat{u}_i : i \in \mathcal{I}_s\}$. By exchangeability of the calibration points and the fresh draw $(X, Y, S)$ within subgroup $s$, the rank

$$R_s := \sum_{i \in \mathcal{I}_s} \mathbf{1}\{\widehat{u}_i \leq \widehat{u}(Y, X, s)\}$$

is discrete uniform on $\{0, 1, \ldots, N_s\}$. Consequently,

$$\widehat{F}_s\big(\widehat{u}(Y, X, s)\big) = \frac{R_s}{N_s}$$

takes values on the grid $\{0, 1/N_s, \ldots, 1\}$, and

$$P\Big(\widehat{F}_s(\widehat{u}) \leq t \mid S = s\Big) = \frac{\lfloor tN_s \rfloor + 1}{N_s + 1} \in \left[ \frac{\lfloor tN_s \rfloor}{N_s + 1}, \frac{\lfloor tN_s \rfloor + 1}{N_s + 1} \right].$$

An analogous statement holds for subgroup $s'$ with $N_{s'}$.

**Grid-mismatch bound.** For any $t \in [0, 1]$, both distribution functions lie on grids with mesh sizes $(N_s + 1)^{-1}$ and $(N_{s'} + 1)^{-1}$, respectively. Hence

$$\Big| P\big(\widehat{F}_s(\widehat{u}) \leq t \mid S = s\big) - P\big(\widehat{F}_{s'}(\widehat{u}) \leq t \mid S = s'\big) \Big| \;\leq\; \frac{1}{\min\{N_s, N_{s'}\} + 1}.$$

Taking the supremum over $t$ gives the claimed Kolmogorov bound. $\qquad\square$

### A.4 ADDITIONAL EXPERIMENTAL DETAILS

**Datasets** We consider a total of 6 datasets from previous studies with two-dimensional outcomes. Specifically, we include 3 datasets (births1, air and wage) from Cevid et al. (2022), 1 dataset (energy) from Wang et al. (2024), 1 dataset (households) from Camehl et al. (2024), and 1 dataset (house) from Feldman et al. (2023). The data preprocessing follows the setup described in Grinsztajn et al. (2022). Table 2 provides the detailed characteristics of each dataset.

Table 2: Summary of datasets considered in this study.

| Source | Dataset | Sample size | Outcome $p$ | Covariate $d$ | Subgroup level $K$ |
|---|---|---|---|---|---|
| Cevid et al. (2022) | births1 | 10,000 | 2 | 23 | 8 |
| | air | 10,000 | 6 | 15 | 7 |
| | wage | 10,000 | 3 | 77 | 2 |
| Wang et al. (2024) | energy | 768 | 2 | 5 | 6 |
| Camehl et al. (2024) | households | 7,207 | 2 | 16 | 4 |
| Feldman et al. (2023) | house | 21,613 | 2 | 16 | 3 |

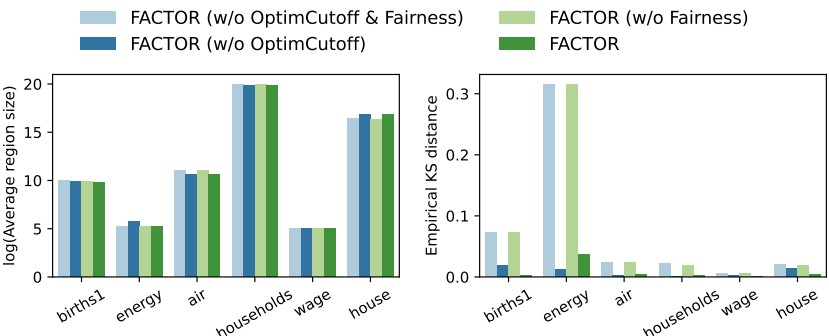

Figure 3: Ablation study on real-world datasets

**Ablation study** Figure 3 presents an ablation study evaluating the contribution of each component in FACTOR. Averaged over the six real-world benchmarks, we obtain: (1) FACTOR without both OptimCutoff and the fairness module; (2) FACTOR without OptimCutoff; (3) FACTOR without the fairness module; and (4) the full FACTOR. The corresponding results are: log(AvgSize) of 11.27, 11.32, 11.25, and 11.23, and KS distance of 0.08, 0.01, 0.08, and 0.01, respectively. These results show that the fairness module does not significantly increase region size when paired with the optimized cutoff, and that the fairness step reduces KS distance to its lowest levels while maintaining competitive region sizes.

**Minimum subgroup coverage** Beyond KS distance, we also report the p%-rule (minimum subgroup coverage), where higher values indicate better fairness across groups. As shown in Figure 4, FACTOR achieves values consistently closest to 100% on both synthetic and real datasets, demonstrating that it provides the strongest subgroup coverage among all compared methods.

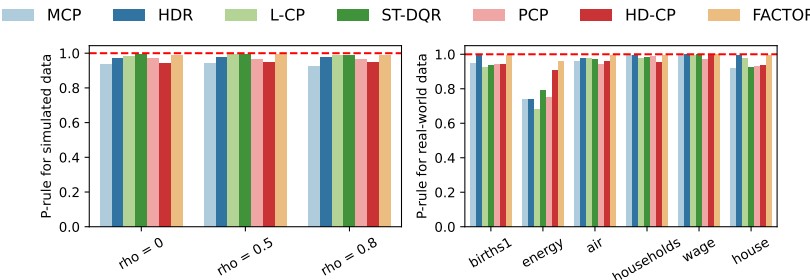

Figure 4: Average p% rule for the calibration set

