# OpenReview forum: "FACTOR: Fairness-Aligned Conformal Transport for Multivariate Mixed Outcomes"
_ICLR.cc/2026/Conference — Submitted to ICLR 2026_

### Official Review · Reviewer_vD5Z · 2025-10-29

**Soundness:** 4
**Presentation:** 4
**Contribution:** 4
**Rating:** 8
**Confidence:** 2

**Summary:**

The paper introduces FACTOR, a framework for constructing fair, compact, and distribution-free conformal prediction regions for multivariate mixed-type outcomes. It combines optimal transport (OT), normalizing flows, and fairness synchronization to ensure both marginal and subgroup-conditional coverage guarantees and was compared against several baselines. The method provides finite-sample fairness bounds, $ O(1/N) $, efficient region construction, and interpretable visualization capabilities.

**Strengths:**

- The paper introduces a novel framework that combines CP, OT, and fairness calibration in an elegant manner.
- FACTOR is presented with good theoretical rigor, with proofs, and has a solid mathematical foundation
- FACTOR is evaluated on both synthetic and (6) real-world datasets, resulting in a good empirical evaluation.
- FACTOR achieves a good balance of conformal efficiency (prediction region size) while also achieving the subgroup coverage parity. All without any additional costs to time elapsed.
- The paper was well presented and was clear when presenting the methodology

**Weaknesses:**

- It would be nice to see results with higher-dimensional outcomes, especially for a scalability analysis. The experiments are limited to low-dimensional outcomes. Would higher dimensions be feasible, or would we need low-rank approximations?
- While it is mentioned that FACTOR can be extended to other fairness definitions, it would be nice to see some results for a metric other than demographic parity.

**Questions:**

See weaknesses

- Are the interpretable visualizations referring to Figures 1 and 2? If so, could you justify how a practitioner would use these to ease analysis more so than any other plot?
- How does your method scale with calibration size? Are there areas where parallelism can be applied?


- (Clarification) In Lemma 1, is $\hat{v}{(1)}$ supposed to be $0$, or is there an implicit definition of $\hat{v}{(0)} = 0$?
The “Why it works” section suggests that $\hat{v}{(1)} = 0$ when it refers to the one-sided rule as corresponding to the interval $[\hat{v}{(1)}, \hat{v}_{(\hat{k})}]$.

---

> ### Author Response · Authors · 2025-11-25
>
> 1. **Higher-dimensional outcomes and scalability**
>
> We agree that evaluating FACTOR for higher $p$ is important. In the revision, we include an additional experiment using the "Air" data with a 6-dimensional outcome vector (max_PM2.5, max_NO2, max_O3, max_PM10, max_CO, max_SO2), using “Weekday’’ as the protected variable. We compare FACTOR with MCP, HDR, L-CP, ST-DQR, PCP, and HD-CP (the last three methods are newly added as suggested by other reviewers). The [[full result is presented here]](https://anonymous.4open.science/r/ICLR2026-13252-DEFC/real_data/real_data_example_p=6.pdf). FACTOR performs well in this higher-dimensional setting:
> - **Second-smallest region size**, only slightly larger than HDR,
> - **Smallest KS distance** (best subgroup calibration) among all methods,
> - **Second-shortest runtime** (only slightly slower than L-CP, which does not enforce any fairness constraint),
> - No optimization instabilities relative to $p = 2$.
>
> These results suggest that FACTOR scales reasonably to higher-dimensional responses with the current architecture. We also note in the revision that for substantially larger $p$, low-rank ICNN parameterizations or structured flows would be natural directions for more parameter-efficient OT maps.
>
> 2. **Other fairness notions beyond demographic parity**
>
> We agree that evaluating additional fairness metrics is valuable. In the revision, we report the p%-rule (the minimum subgroup coverage ratio), where values closer to 100% indicate better fairness across groups. We include the [[p%-rule for both synthetic and real datasets here]](https://anonymous.4open.science/r/ICLR2026-13252-DEFC/p_rule_fairness.pdf), and FACTOR consistently scores closest to one among all methods. We also briefly discuss how FACTOR could incorporate other fairness criteria (e.g., equalized odds applied to conditional coverage events), but we avoid overclaiming beyond what we evaluate empirically.
>
> 3. **Clarification of Figures 1 and 2 (interpretable visualizations)**
>
> Yes, the interpretable visualizations refer to Figures 1 and 2. To clarify their value for practitioners, we add a discussion explaining that the plots allow users to:
> - Inspect how multiple outcomes interact jointly (e.g., tradeoffs between adverse events, spatially correlated pollutants, joint clinical biomarkers).
> - Understand how region geometry changes across subgroups, which is not visible in marginal or univariate summaries.
> - Compare how alternative methods behave for the same conditional X, highlighting differences in calibration or conservativeness.
>
> We emphasize that these plots are not required for using FACTOR but provide an intuitive window into multivariate coverage behavior.
>
> 4. **Scaling with calibration size and parallelism**
>
> In our experiments, FACTOR runs efficiently on calibration sets ranging from 768 to 21,613 observations. The elapsed times reported in Figures 1 and 2 include the calibration step for the entire calibration points. Because FACTOR’s calibration step is based on evaluating the OT map on calibration points and scanning over latent scores, the method is linearithmic in calibration size and trivially parallelizable across calibration samples. We clarify this in the revision.
>
> 5. **Lemma 1 clarification**
>
> Yes, you are right, and thank you for pointing this out. We have revised Lemma 1 to ensure that the mapping and the interval notation are explicit and consistent with the “Why it works’’ section.

---

> > ### Comment · Reviewer_vD5Z · 2025-11-25
> >
> > Thanks for addressing my questions and concerns. I will be maintaining my score.

---

> > > ### Author Response · Authors · 2025-11-26
> > > **Thanks**
> > >
> > > Dear Reviewer, thank you very much for your positive feedback on our rebuttals! We are glad to hear that our clarifications addressed your concerns. Please feel free to let us know if any additional questions arise. Thanks again for your insightful comments and for your hard work on our manuscript! - Authors

---

### Official Review · Reviewer_aMJP · 2025-10-31

**Soundness:** 2
**Presentation:** 4
**Contribution:** 2
**Rating:** 4
**Confidence:** 3

**Summary:**

The paper proposes an approach to create compact and equitable prediction regions in multivariate response settings. They propose learning the optimal-transport map in a latent space via normalizing flows with input convex neural networks, transforming the multivariate problem into a univariate one that can be used to construct the conformal region. They also propose a sliding window trick to get compact prediction regions. Finally, they propose synchronizing the latent-space ranks of specific subgroups to achieve a finite-sample coverage bound for each subgroup, enabling equitable prediction regions. They perform experiments on both synthetic and real-world datasets (all with a bivariate outcome), which showcase promising results, certainly regarding the subgroup guarantees.

**Strengths:**

- The approach to get an equitable set is very clean, well-motivated, and demonstrated to work on real-world data.
- The approach is straightforward and presented well.

**Weaknesses:**

- For me, it is not so clear what the research deltas are compared with the works of Thurin et al. (2025) and Klein et al. (2025). It seems that both these works introduce a similar procedure as you, besides the sub-group guarantee and the volume optimality, if so, I think this should be more clearly stated in the methodology section.
- The papers claim to present an approach for mixed multivariate responses. However, only bivariate responses are tested, and it seems that only continuous responses are evaluated.
- One thing I notice is that in the discussion and also in the related work, you skip over the high-density approaches, which, I intuitively find, provide very good prediction regions. A more elaborate discussion could improve the work. Examples of such high-density works are:
    - Dheur, V., Bosser, T., Izbicki, R. and Ben Taieb, S., 2024. Distribution-free conformal joint prediction regions for neural marked temporal point processes. *Machine Learning*, *113*(9), pp.7055-7102.
    - Izbicki, R., Shimizu, G. and Stern, R.B., 2022. Cd-split and hpd-split: Efficient conformal regions in high dimensions. *Journal of Machine Learning Research*, *23*(87), pp.1-32.
    - Jonkers, J., Coopman, F., Duchateau, L., Van Wallendael, G. and Van Hoecke, S., 2025. Reliable uncertainty quantification for 2D/3D anatomical landmark localization using multi-output conformal prediction. *arXiv preprint arXiv:2503.14106*.

**Questions:**

- In Fig. 1. Your example regions are quite similar for HDR, L-CP, and FACTOR; however, the KS distance is quite different. Can you elaborate on that?
- What is the reason for in the synthetic datasets that the efficiency is best out of all approaches and for the real world experiments is worst (disregarding MCP)?
- So by enforcing the fairness, you get a suboptimal solution, in terms of transport map, what is the effect on efficiency after the calibration procedure?
- Regarding Eq. 5, it is not clear to me what r_{\alpha, 1} and r_{\alpha, 2} represent here; you should explain it a bit more clearly, in my opinion.
- Could you maybe explain a bit more thoroughly why that optimization in the rank space results in the shortest region in the outcome space?
- Regarding the actual creation of the prediction region,  do we need to do a grid search over the response domain to approximate the prediction region?

---

> ### Author Response · Authors · 2025-11-25
>
> 1. **Research deltas vs. Thurin et al. (2025) and Klein et al. (2025)**
>
> We agree that the differences with Thurin et al. (2025) and Klein et al. (2025) should be stated more clearly in the methodology section. In the revision, we have now added a dedicated paragraph that contrasts these approaches with FACTOR. In brief, Thurin et al. (2025) develop an OT-based conformal framework that uses transport to define multivariate scores, but they focus on marginal coverage and do not address subgroup-conditional guarantees or an explicit fairness mechanism. Klein et al. (2025) use conformalized Gaussian scoring to construct multivariate regions, relying on Gaussian approximation to define a score with good geometric properties; again, the method targets global efficiency but not subgroup calibration or fairness.
> FACTOR differs in three main ways:
> * It uses an ICNN-based OT map to construct a rank that is directly tied to optimal transport and convex function.
> * It introduces a functional synchronization step that enforces finite-sample subgroup-conditional coverage, addressing fairness directly.
> * It includes a volume-optimization step (revised in the new version, see below) designed to minimize estimated region volume in outcome space, not just in the latent rank space.
>
> We now emphasize these deltas explicitly in the methodology and related work sections.
>
> We have also greatly expanded our related work section, thanks to this reviewer and Reviewer wdLA. We reproduce this updated section in a comment below.
>
> 2. **Higher dimensions (p > 2) and mixed responses**
>
> We agree that it is important to move beyond purely bivariate continuous outcomes and to demonstrate behavior in higher dimensions.
>
> - **Mixed responses**: In the [[updated results for real data here]](https://anonymous.4open.science/r/ICLR2026-13252-DEFC/real_data/real_data.pdf), we clarify that the house ($p=2$, one continuous + one discrete outcomes) and wage datasets ($p=3$, two continuous + one discrete outcomes) have already been used to illustrate how mixed-type responses can be handled in practice and how FACTOR embeds them into the outcome space.
>
> - **New p = 6 experiment**: We also [[added a new experiment here]](https://anonymous.4open.science/r/ICLR2026-13252-DEFC/real_data/real_data_example_p=6.pdf) on the "Air" data with a 6-dimensional response: max_PM2.5, max_NO2, max_O3, max_PM10, max_CO, max_SO2 using “Weekday” as the protected variable. We compare FACTOR with L-CP, HDR, MCP, ST-DQR, PCP, and HD-CP (the last three methods are newly added as suggested by other reviewers).
> In this $p = 6$ setting, FACTOR achieves:
>   - Second-smallest region volume, only slightly larger than HDR,
>   - Smallest KS distance (best subgroup calibration) among all methods,
>   - Second-shortest runtime (only slightly slower than L-CP, which does not enforce any fairness constraint),
>   - No optimization instabilities relative to $p = 2$.
>
> These results, now included in the paper, show that FACTOR scales reasonably to higher-dimensional outcomes while retaining good efficiency and fairness.
>
> 3. **High-density approaches in the literature review**
>
> We appreciate the pointer to high-density (HDR) and related methods, which indeed provide strong baselines for compact prediction regions. In the revised related work section, we now explicitly discuss:
> - HDR-style regions and high-density conformal regions (e.g., Dheur et al., 2024; Izbicki et al., 2022),
> - Recent multi-output conformal methods for landmark localization (Jonkers et al., 2025).
>
> We also include HD-CP (a high-density extension of PCP) in our experimental comparisons, alongside ST-DQR and PCP, so that FACTOR is evaluated against a broader set of state-of-the-art methods for compact regions; see [[results for the simulated data]](https://anonymous.4open.science/r/ICLR2026-13252-DEFC/sim/sim_n5000.pdf) and [[results for the real-world data]](https://anonymous.4open.science/r/ICLR2026-13252-DEFC/real_data/real_data.pdf).
>
> 4. **Fig. 1: similar regions but different KS distances**
>
> Regarding Fig. 1, we agree that the example regions can appear quite similar for a single covariate value. However, the KS distance we report is computed across the calibration set with a sample size of 2000, averaging over many different values of $X$. Thus, even if HDR, L-CP, and FACTOR yield visually similar regions for one specific $X$, their behavior can differ substantially for other values of $X$. This leads to noticeable differences in the empirical KS distances and average region volumes when aggregated over the calibration distribution. We have clarified this in the figure caption and text.

---

> ### Author Response · Authors · 2025-11-25
>
> 5. **Synthetic vs. real-world efficiency and revised volume optimization**
>
> The question was raised as to why FACTOR is most efficient on synthetic data but can be less efficient on some real-world datasets, and how optimization in the rank space relates to outcome-space volume. While there is no guarantee that we can achieve the best of all approaches across every real-world experiment while ensuring fairness on conditional coverage, it is guaranteed that we achieve the desired coverage within the class of fair functions that are **as close as possible (measured by L2 loss)** to the conformal scores $u(Y,X,S)$ produced by the OT map.
> Regarding optimization in the rank space and its relationship to outcome-space volume, we note that following the recommendation of Reviewer `wdLA`, we see that a monotone transformation alone does not guarantee that shorter intervals in latent space correspond to smaller sets in outcome space. To address this, we revised Remark 3 and updated equation (6) to actually optimize volume in the outcome space:
> $$C_{\alpha}^{opt}(X,S)=\arg\min_{C_{\alpha}}\[|C_{\alpha}|: P(Y\in C_{\alpha}(X,S))\geq 1-\alpha\]=\arg\min_{r_{\alpha,1}, r_{\alpha,2}}\[|C_{\alpha}|: P(r_{\alpha,1}\leq v^{*}(Y,X,S)\leq r_{\alpha,2})\geq 1-\alpha\]$$
>
> To compute the region size, we run importance sampling using the calibration dataset $I_3$:
> $$|C_\alpha|
> = \int 1(y \in C_{\alpha}(X,S)) dy
> = E_{Y \sim p_Y(y \mid X,S)}
> \left[
> \frac{1(Y \in C(X,S))}{p_{Y}(Y \mid X, S)}
> \right]
> \approx
> \frac{1}{|I_3|}
> \sum_{i \in I_3}
> \frac{1(r_{\alpha,1} \le v^{*}(Y_i, X_i, S_i) \le r_{\alpha,2})}{p_Y(y_i \mid X_i, S_i)}.$$
> Here, each conformal score is inversely weighted by the density estimator for the outcome. This is consistent with equation (130) for ``A Unified Comparative Study with Generalized Conformity Scores for Multi-Output Conformal Regression''. The intuition is that the interval $[r_{\alpha,1}, r_{\alpha,2}]$ achieving the desired $\alpha$-coverage should contain as few outcomes as possible, where each outcome is weighted inversely by its probability, thereby minimizing the volume size in the outcome space. Algorithm 1 will be updated accordingly.
>
> In more detail, we use **importance sampling** to obtain the estimates of the prediction region, and we run a **grid search** for each dimension of the outcome to create the boundary of the prediction region.
>
> 6. **Effect of fairness on efficiency (ablation study)**
>
> We agree that it is important to quantify how much efficiency is lost when imposing fairness via synchronization. In the revised paper, we add an ablation study that compares:
> - FACTOR (without optimized cutoff and without fairness),
> - FACTOR (without optimized cutoff),
> - FACTOR (without fairness),
> - FACTOR (with both optimized cutoff and fairness — our proposed method).
>
> Averaged over the six real-world benchmarks, we obtain
> |                    | FACTOR (w/o OptimCutoff & Fairness) | FACTOR (w/o OptimCutoff) | FACTOR (w/o Fairness) | FACTOR |
> |--------------------|--------------------------------------|---------------------------|-------------------------|--------|
> | log(AvgSize)       | 11.27                                | 11.32                     | 11.25                  | **11.23** |
> | KS Distance        | 0.08                                 | 0.01                      | 0.08                   | **0.01** |
>
> These results show that:
> - Enforcing fairness does not substantially worsen region size when combined with the new optimized cutoff, and
> - The fairness step reduces KS distance to the lowest levels while keeping region size competitive or better than the unfair variants.
>
> We will include this table and [[full results for the 6 benchmark datasets]](https://anonymous.4open.science/r/ICLR2026-13252-DEFC/real_data/ablation_study.pdf) in the revised paper.
>
> 7. **Clarifying $r_{\alpha,1}$ and $r_{\alpha,2}$ in Equation (5)**
>
> Thank you for pointing out the ambiguity. In the revised version, we state that $r_{\alpha,1}$ and $r_{\alpha,2}$​ are cutoff values on the fair latent scores $v(Y,X,S)$, where $v(\cdot)$ denotes the synchronized transported distance. For any new $(X,S)$, the prediction region $C_\alpha(X,S)$ is defined as the set of all $y$ such that $r_{\alpha,1} \leq v(y,X,S) \leq r_{\alpha,2}$. These cutoffs are chosen to ensure correct coverage and, after the revision, to minimize the estimated outcome-space volume via the **importance-sampling scheme**.

---

> ### Author Response · Authors · 2025-11-26
> **Expanded Literature Review (1/2)**
>
> Thanks to this reviewer and reviewer aMJP, we've greatly expanded our "Related Works" section, which we reproduce here.
>
> **Shape-constrained methods.**
> The earliest extensions of conformal prediction for univariate regression to multivariate regression consisted of constructing Cartesian products of marginal prediction intervals, producing hyperrectangles that did not account for correlations among outcome variables and were overly conservative (Neeven, 2018). Ellipsoidal prediction intervals were proposed to incorporate covariance information and produce smaller prediction sets under certain conditions, but they were restricted to convex geometric shapes assuming an underlying elliptical structure and were unable to capture more flexible distributions (Johnstone, 2021; Messoudi, 2022).
>
> **Copula-based methods.**
> To avoid fixed geometric assumptions, simple parametric copulas have been shown to work for certain datasets (Messoudi, 2021). Vine copulas have been proposed to avoid strong parametric assumptions and to directly model dependencies in the outcome distribution (Park, 2024), but loss of coverage can occur when the estimated copula of the conformal scores deviates from the true copula, and finite-sample validity cannot be guaranteed (Dheur, 2025).
>
> **Volume-minimizing and high-density region methods.**
> Seeking to minimize volume, Tumu et al. (2024) restrict prediction regions to convex shapes, using heuristic clustering algorithms to adaptively partition the data and maintain coverage. More flexible strategies have been proposed that optimize prediction regions over arbitrary norms, thereby removing restrictive convexity constraints while still achieving exact finite-sample coverage (Braun, 2025). However, reliance on first-order optimization techniques introduces the risk of convergence to poor local minima, and aggressive volume reduction can compromise conditional coverage across subgroups. A related line of work focuses on high-density regions (HDR), which define prediction regions as estimated density level sets (Izbicki, 2022; Dheur, 2024; Jonkers, 2025). These methods can produce relatively efficient regions, but they rely on accurate density estimation and do not provide exact finite-sample coverage in the presence of mixed outcomes.

---

> ### Author Response · Authors · 2025-11-26
> **Expanded Literature Review (2/2)**
>
> **Latent-space quantile methods.**
> A recent approach is to first map the conditional distribution of the response into a latent space where the level sets of the density are convex, and then transform these sets back into the original space. This can be achieved using a deep generative model that learns a latent representation of the response with an approximately unimodal distribution, e.g., using directional quantile regression and conditional variational autoencoders (CVAE). The spherically transformed directional quantile regression (ST-DQR) method of Feldman et al. (2023) can produce smaller prediction regions, but a potential limitation is that its performance depends heavily on the quality of the CVAE, which can be improved by incorporating more modern generative models such as normalizing flows (Kobyzev, 2020). Related probabilistic generative approaches fit a conditional generative model for the outcome and construct prediction sets by retaining sampled responses with the largest estimated densities (Wang, 2023).
>
>
> **Optimal transport methods.**
> Our work is most similar to recent optimal transport (OT) methods, which seek to define a meaningful ordering in multidimensional spaces (Chernozhukov, 2024; Hallin, 2021; Hallin, 2023). Thurin et al. (2025) and Klein et al. (2025) extended conformal inference techniques to multivariate conformal score functions by transporting the response distribution to a uniform reference measure using an OT map. Computationally, Klein et al. (2025) uses general entropic maps and establishes finite-sample coverage guarantees. Although these methods introduce OT-based scores, they focus exclusively on marginal coverage, do not incorporate mechanisms for subgroup-conditional guarantees, are not tailored to mixed discrete–continuous outcomes, and do not optimize region volume in the outcome space.
>
> A related line of work learns a transformation into a simple reference distribution using normalizing flows. CONTRA (Fang, 2025) uses a real-valued non-volume preserving (RealNVP) bijective flow (Dinh, 2017) to push the response toward a Gaussian reference distribution and defines the conformity score as the Euclidean distance to the origin in the transformed space. A variant of CONTRA, called ResCONTRA, attempts to improve predictive performance by training a second normalizing flow on the residuals. However, this approach is less computationally efficient because it requires fitting two complex models on the same dataset. In contrast, our method trains a single flow and additionally introduces a functional synchronization step to guarantee subgroup-conditional coverage for fairness, a feature that is not addressed in CONTRA. The method we develop also transforms the response into a simpler latent space using the gradient of an Input Convex Neural Network (ICNN) to approximate the OT map, which has been shown to provide universal approximation guarantees for convex functions (Chen, 2019) and their gradients (Huang, 2020).

---

### Official Review · Reviewer_wdLA · 2025-10-31

**Soundness:** 2
**Presentation:** 1
**Contribution:** 3
**Rating:** 4
**Confidence:** 3

**Summary:**

The paper targets an important problem, about mixed type multi-dimensional outcome with applied examples.

The proposed method FACTOR hinges on transforming the conditional distribution of Y given X (and S) into Uniform reference distributions via normalizing flow (NF). The NF is learned to optimize the KL distance of the transformed distribution to the reference, and this optimaization problem is approximately solved using solutions in a space that has simpler function forms.

Some nice features of FACTOR includes that it synchronizes latent-space ranks across subgroups, to achieve distribution-free group-wise coverage.  Empirical studies on synthetic and six real-world datasets show that FACTOR can achieve target coverage with smaller prediction regions than the competitors presented.

However, the current draft missed noticing and comparing to a couple of relevant alternative methods for multi-target CP.

I have major concern about the presentation. While the overall idea and theory are very interesting, the writing makes it difficult and frustrating to follow and check the details. This should be fixable though!

**Strengths:**

The idea of transforming y|x to a standard distribution is a powerful backbone of this work.
The way of learning that transformation by posing this as an optimization problem (1) and approximating the solution (3, 4, Thm 2) seems very neat.

It’s great to see that the algorithm, which involves multiple nontrivial steps, has been carefully designed and implemented.

**Weaknesses:**

Some literatures not discussed in "Related work" that also allow flexible shape multi-dimensional CP are as follows. These works could also serve as useful benchmarks for comparison with the proposed FACTOR method in empirical studies.
PCP by Wang et al (Probabilistic conformal prediction using conditional random samples) AISTATS, 2023.
CONTRA by Fang et al (Conformal Prediction Region via Normalizing Flow Transformation) ICLR, 2025.

Presentation-wise, there were often missing transitions between the bullet-point style paragraphs, and there are quite a lot of ambiguous statements. The draft would benefit from a more careful round of proofreading. E.g., the authors could inspect it as if they were first-time readers.

Quote Remark 3"Because prediction regions in outcome space are inverse
images of these intervals under a monotone transport, shorter intervals lead directly to smaller,
more efficient prediction regions." I don't agree with this statement. As a simple counter example, v=u^3 is a monotone mapping, but shorter intervals in u, does not necessarily lead to shorter invervals in v. You can see that intervcal of size 0.5 in u can lead to intervals of size anywhere from 1/8 to 7/8 in v. It's the magnitude of the change of rate (slope) of the mapping at different u that counts, not just that the slope is always positive (monotonely increasing).
Because of this disagreement, I have some doubt on the cutoff optimization strategy in section 2.3. (Maybe I misunderstood something here, please let me know.)


Quote Remark 1 "In the univariate case, u∗(·) reduces to the CDF F(·), so the OT-based rank is a direct
multivariate generalization of quantiles." Here, by u^*(.) do you mean a function y while fixing X and S? And what is "the cdf F"? The meaning of this sentence is unclear. In general, when a function has multiple input/argument, please state more clearly which argument is a variable and what others you are fixing/conditioning upon when you call it a function. This issue occurs at least in one other place, where function u(.) needs to be projected into a space of v(.) functions in sec 2.2.

Also, read Theorem 2 again. It's not tight and clear mathematically. Basic setups/context and assumptions need to be stated INSIDE the Theorem.
Actually, for the space G defined in L165, is it for a fixed X or what? The general idea and theory sound very cool but the presentation makes it quite frustrating when I tried to undersatnd the details.

**Questions:**

L128 What is a "protected subgroup label"? (I can guess what it means after reading a few more pages, but I believe this should be explained explicitly upfront. Perhaps present an applied example in the beginning.)

If some dimension of Y is nominal categorical, that is, the ordering of the category is meaningless, is it still proper to use your framework of \mathcal{Y} \subset R^p?

There seems to be great similarity between the idea of FACTOR and CONTRA. Note that FACTOR seeks a transformation (ended up using normalizing flow) of y|x into random vector that follows the Uniform distribution on a p-dimensional ball; CONTRA learns a normalizing flow that transforms y|x into a p-dimensional standard normal random vector. Both methods then define the non-conformity score as distance of transformed variable to the origin and obtained empirical quantile to construct conformal regions for new data points.
I think a key difference is that in learning the NF, FACTOR tried to minimize the KL divergence between transformed and reference distributions, while CONTRA tried to minimize negative log likelihood. Both ideas make sense, and it would be interesting to see how they compare in empirical studies of various kinds.

---

> ### Author Response · Authors · 2025-11-24
>
> 1.  **Concern about monotone transport and interval length**
>
> We thank the reviewer for this excellent catch. We agree that the original statement in Remark 3 needs to be updated. As the reviewer pointed out, a monotone transformation alone does not guarantee that shorter intervals in latent space correspond to smaller sets in outcome space. To address this, we revised Remark 3 and updated equation (6) to actually optimize volume in the outcome space:
> $$C_{\alpha}^{opt}(X,S)=\arg\min_{C_{\alpha}}\[|C_{\alpha}|: P(Y\in C_{\alpha}(X,S))\geq 1-\alpha\]=\arg\min_{r_{\alpha,1}, r_{\alpha,2}}\[|C_{\alpha}|: P(r_{\alpha,1}\leq v^{*}(Y,X,S)\leq r_{\alpha,2})\geq 1-\alpha\]$$
>
> To compute the region size, we run importance sampling using the calibration dataset $I_3$:
> $$|C_\alpha|
> = \int 1(y \in C_{\alpha}(X,S)) dy
> = E_{Y \sim p_Y(y \mid X,S)}
> \left[
> \frac{1(Y \in C(X,S))}{p_{Y}(Y \mid X, S)}
> \right]
> \approx
> \frac{1}{|I_3|}
> \sum_{i \in I_3}
> \frac{1(r_{\alpha,1} \le v^{*}(Y_i, X_i, S_i) \le r_{\alpha,2})}{p_Y(y_i \mid X_i, S_i)}.$$
> Here each conformal score is inversely weighted by the density estimator for the outcome. This is consistent with equation (130) for ``A Unified Comparative Study with Generalized Conformity Scores for Multi-Output Conformal Regression''. The intuition is that the interval $[r_{\alpha,1}, r_{\alpha,2}]$ achieving the desired $\alpha$-coverage should contain as few outcomes as possible, where each outcome is weighted inversely by its probability, thereby minimizing the volume size in the outcome space. Algorithm 1 will be updated accordingly.
>
> 2. **Missing related work (PCP, CONTRA) and new experiments**
>
> Thank you for pointing out these important references. We have substantially expanded the Related Work section to include PCP (Wang et al., 2023), CONTRA (Fang et al., ICLR 2025), and high-density approaches. For PCP, we were able to obtain code and have now included PCP as a comparator in all of our synthetic and real data experiments. For CONTRA, since public code is not available, we could not implement it, but we now summarize similarities and key differences in terms of modeling choices and training objectives. Further, following other reviewers’ suggestions, we have added three additional comparators:
> * Transformed directional quantile regression (ST-DQR): Feldman et al., 2023
> * Probabilistic conformal prediction (PCP): Wang et al., 2023
> * High-density CP (HD-CP): Wang et al., 2023b, extension to PCP by retaining the samples with the highest density.
>
> The [[updated empirical results for the simulations are presented here]](https://anonymous.4open.science/r/ICLR2026-13252-DEFC/sim/sim_n5000.pdf), which show that FACTOR achieves the **smallest region size**, the **smallest KS distance** (best subgroup calibration), and the **second shortest runtime** (slower than L-CP, which does not enforce the fairness constraint).
>
> We also [[updated the 6 real-world dataset experiments here]](https://anonymous.4open.science/r/ICLR2026-13252-DEFC/real_data/real_data.pdf). The average metrics across these methods presented in the table below indicate FACTOR has the **smallest region volume** and the **smallest KS distance**.
>
> |               | MCP   | HDR   | L-CP  | ST-DQR | PCP   | HD-CP | FACTOR |
> |-------------- |-------|-------|-------|--------|-------|-------|--------|
> | log(AvgSize)  | 14.12 | 11.56 | 11.27 | 14.13  | 14.12 | 14.10 | **11.23** |
> | KS Distance   | 0.07  | 0.05  | 0.08  | 0.06   | 0.08  | 0.05  | **0.01** |

---

> ### Author Response · Authors · 2025-11-24
>
> 3. **Clarifying similarities and differences between FACTOR and CONTRA**
>
> We now include a dedicated comparison in the Related Work section. In summary: Both FACTOR and CONTRA learn a normalizing flow to transform $Y \mid X$ into a simple reference distribution and define the score as the distance to the origin. CONTRA relies on RealNVP (Dinh et al., 2016) to learn a bijective transformation directly, whereas FACTOR uses the gradient of an ICNN to approximate the optimal-transport map. The ICNN-based approach is supported by theoretical results showing universal approximation of convex functions (Chen et al., 2019) and their gradients (Huang et al., 2021), which provides additional structure for learning transport maps. A variant of CONTRA, called ResCONTRA, attempts to improve predictive performance by training a second normalizing flow on the residuals. However, this approach is less computationally efficient because it requires fitting two complex models on the same dataset. In contrast, FACTOR trains a single flow and additionally introduces a functional synchronization step to guarantee subgroup-conditional coverage for fairness, a feature that is not addressed in CONTRA. The loss functions considered are similar, as there is a connection between the MLE and the KL divergence, as pointed out in the paper “Multivariate Quantile Function Forecaster”. The main differences are summarized in the table below:
>
> | Aspect                | CONTRA                          | FACTOR                                                             |
> |---------------------- |--------------------------------- |------------------------------------------------------------------- |
> | Reference distribution| $N(0, I_p)$                     | Uniform on unit ball                                               |
> | Architecture          | RealNVP (Dinh et al., 2016)      | Gradient of ICNN (universal approximation under convexity)         |
> | Fairness              | No fairness mechanism            | Functional synchronization ensuring subgroup coverage               |
> | Extensions            | ResCONTRA trains two flows (slower) | FACTOR uses a single flow                                        |
>
>
> 4. **Improved presentation and clarity**
>
> * We agree that our previous notation was not clear enough. In the revision, we now explicitly treat $Y$ as the free variable and $(X,S)$ as conditioned inputs. The OT map is now written as $q_{Y \mid X,S}(Y \mid X,S)$ to make the dependence explicit. In remark 1, we clarify that for fixed $X$ and $S$, the univariate OT map reduces to the conditional CDF. We also revised Section 2.2 to clarify projection from $u(\cdot)$ into the function space $\mathcal{G}$.
>
> * We now define “protected subgroup label” explicitly when first introduced. A protected subgroup label refers to the value of the sensitive attribute $S$ (e.g., race, gender, socioeconomic group). We also introduce an applied example early in Section 1 to make this concrete.
>
> * If some dimension of $Y$ is nominal categorical with no meaningful ordering of the outcomes, we can still apply our method, e.g., running our algorithm within each outcome group. To remove any confusion regarding the outcome dimension and how our method would apply, we explain this and modify the outcome space to be written as $\mathcal{Y}$.
>
> * In the revision, we have substantially improved the presentation for clarity and readability. Specifically, we expanded the Related Work section to incorporate the suggested references (PCP, CONTRA, and high-density approaches), added missing contextual transitions, and rewrote several bullet-style segments into smoother prose with clearer explanations and notation. We also added remarks throughout Sections 2 and 3 to guide readers and remove earlier ambiguities.

---

> ### Author Response · Authors · 2025-11-26
> **Expanded Literature Review (1/2)**
>
> Thanks to this reviewer and reviewer aMJP, we've greatly expanded our "Related Works" section, which we reproduce here.
>
> **Shape-constrained methods.**
> The earliest extensions of conformal prediction for univariate regression to multivariate regression consisted of constructing Cartesian products of marginal prediction intervals, producing hyperrectangles that did not account for correlations among outcome variables and were overly conservative (Neeven, 2018). Ellipsoidal prediction intervals were proposed to incorporate covariance information and produce smaller prediction sets under certain conditions, but they were restricted to convex geometric shapes assuming an underlying elliptical structure and were unable to capture more flexible distributions (Johnstone, 2021; Messoudi, 2022).
>
> **Copula-based methods.**
> To avoid fixed geometric assumptions, simple parametric copulas have been shown to work for certain datasets (Messoudi, 2021). Vine copulas have been proposed to avoid strong parametric assumptions and to directly model dependencies in the outcome distribution (Park, 2024), but loss of coverage can occur when the estimated copula of the conformal scores deviates from the true copula, and finite-sample validity cannot be guaranteed (Dheur, 2025).
>
> **Volume-minimizing and high-density region methods.**
> Seeking to minimize volume, Tumu et al. (2024) restrict prediction regions to convex shapes, using heuristic clustering algorithms to adaptively partition the data and maintain coverage. More flexible strategies have been proposed that optimize prediction regions over arbitrary norms, thereby removing restrictive convexity constraints while still achieving exact finite-sample coverage (Braun, 2025). However, reliance on first-order optimization techniques introduces the risk of convergence to poor local minima, and aggressive volume reduction can compromise conditional coverage across subgroups. A related line of work focuses on high-density regions (HDR), which define prediction regions as estimated density level sets (Izbicki, 2022; Dheur, 2024; Jonkers, 2025). These methods can produce relatively efficient regions, but they rely on accurate density estimation and do not provide exact finite-sample coverage in the presence of mixed outcomes.

---

> ### Author Response · Authors · 2025-11-26
> **Expanded Literature Review (2/2)**
>
> **Latent-space quantile methods.**
> A recent approach is to first map the conditional distribution of the response into a latent space where the level sets of the density are convex, and then transform these sets back into the original space. This can be achieved using a deep generative model that learns a latent representation of the response with an approximately unimodal distribution, e.g., using directional quantile regression and conditional variational autoencoders (CVAE). The spherically transformed directional quantile regression (ST-DQR) method of Feldman et al. (2023) can produce smaller prediction regions, but a potential limitation is that its performance depends heavily on the quality of the CVAE, which can be improved by incorporating more modern generative models such as normalizing flows (Kobyzev, 2020). Related probabilistic generative approaches fit a conditional generative model for the outcome and construct prediction sets by retaining sampled responses with the largest estimated densities (Wang, 2023).
>
>
> **Optimal transport methods.**
> Our work is most similar to recent optimal transport (OT) methods, which seek to define a meaningful ordering in multidimensional spaces (Chernozhukov, 2024; Hallin, 2021; Hallin, 2023). Thurin et al. (2025) and Klein et al. (2025) extended conformal inference techniques to multivariate conformal score functions by transporting the response distribution to a uniform reference measure using an OT map. Computationally, Klein et al. (2025) uses general entropic maps and establishes finite-sample coverage guarantees. Although these methods introduce OT-based scores, they focus exclusively on marginal coverage, do not incorporate mechanisms for subgroup-conditional guarantees, are not tailored to mixed discrete–continuous outcomes, and do not optimize region volume in the outcome space.
>
> A related line of work learns a transformation into a simple reference distribution using normalizing flows. CONTRA (Fang, 2025) uses a real-valued non-volume preserving (RealNVP) bijective flow (Dinh, 2017) to push the response toward a Gaussian reference distribution and defines the conformity score as the Euclidean distance to the origin in the transformed space. A variant of CONTRA, called ResCONTRA, attempts to improve predictive performance by training a second normalizing flow on the residuals. However, this approach is less computationally efficient because it requires fitting two complex models on the same dataset. In contrast, our method trains a single flow and additionally introduces a functional synchronization step to guarantee subgroup-conditional coverage for fairness, a feature that is not addressed in CONTRA. The method we develop also transforms the response into a simpler latent space using the gradient of an Input Convex Neural Network (ICNN) to approximate the OT map, which has been shown to provide universal approximation guarantees for convex functions (Chen, 2019) and their gradients (Huang, 2020).

---

### Official Review · Reviewer_XxR7 · 2025-11-01

**Soundness:** 3
**Presentation:** 4
**Contribution:** 3
**Rating:** 6
**Confidence:** 3

**Summary:**

The paper introduces FACTOR (Fairness-Aligned Conformal Transport for Optimal Regions), a novel framework for constructing fair prediction regions for multivariate, mixed-type (continuous/discrete) outcomes. FACTOR defines a principled multivariate ranking by learning an optimal transport map, approximated by a normalizing flow with input-convex neural networks. This map pushes complex, mixed-type outcomes into a simple uniform latent space, and the norm of the transported point serves as a scalar conformity score, avoiding restrictive geometric assumptions. To ensure equity, FACTOR introduces a functional synchronization step that post-processes the OT-based scores to align their distributions across protected subgroups. This provides a formal, finite-sample O(1/N) bound on the subgroup calibration error, directly addressing a major limitation of many existing conformal methods. The framework includes a sliding-window cutoff optimization procedure that searches for the shortest interval in the rank space to achieve the desired coverage. This technique is shown to produce smaller prediction regions than standard one-sided quantile methods, especially in the presence of finite-sample noise or distributional irregularities.

**Strengths:**

The proposed method combines optimal transport for multivariate ranking, ICNN-based normalizing flows for implementation, a post-hoc synchronization step for fairness, and a sliding-window optimization for efficiency. This framework appears to be novel and well-motivated. The OT formulation provides a sound theoretical basis for a multivariate rank. In particular, a standout contribution is the finite-sample O(1/N_s) bound on subgroup calibration error (Theorem 3). This is a significant step beyond the marginal or asymptotic guarantees offered by most conformal methods and provides a strong theoretical underpinning for the fairness claims.

**Weaknesses:**

The main concern is about the complexity in implementing the proposed method. The method relies on several advanced components (ICNNs, normalizing flows, OT maps). Each component involves a complicated task, a break down in any of these components may lead to unsatisfactory results. For example, the quality of the OT map, and therefore the efficiency of the prediction regions, depends on the underlying normalizing flow model. The paper acknowledges this but does not systematically analyze robustness to model misspecification. Poorly trained flows could lead to distorted ranks and inefficient regions, even if formal coverage is maintained. More theoretical analysis or numerical experiments are needed to analyze the robustness of the method.

**Questions:**

How sensitive is the efficiency of FACTOR's prediction regions to the quality of the learned OT map? For example, if the normalizing flow is poorly trained, how does this affect the region volume compared to baselines?

The experiments are all conducted with p=2. How does the performance (training stability, runtime, region size) of FACTOR change as the dimension of the outcome space p increases?

---

> ### Author Response · Authors · 2025-11-24
>
> 1. We appreciate the reviewer’s question about how the quality of training for the normalizing flow affects the prediction region performance, which is important for any method relying on learned transport maps. We address robustness from both a theoretical and practical perspective.
>
> * **Theoretical perspective**: Although robustness of learned OT maps is an active research area, several useful approximation results exist. Prior work shows that ICNNs can approximate any convex function (Chen et al., 2019) and that their gradients can approximate monotone multivariate maps (Huang et al., 2021). These results imply that, under mild regularity assumptions, an ICNN‑based flow can approximate the optimal OT map universally with small training error. This provides stability for the learned ranking function: small training errors in the flow lead to only small perturbations in the transported ranks, which should have limited impacts on the resulting conformal prediction regions.
> * **Empirical diagnostic and sensitivity**: Thanks to the reviewer’s suggestion, we now propose a simple diagnostic to assess the quality of the learned OT map: we compare the empirical distribution of the transported distance $u(Y,X,S)=\|q(Y,X,S)\|$ on the calibration dataset with the theoretical distribution under the target reference measure, which corresponds to the distribution for the radii of a uniformly distributed vector. The theoretical distribution is approximated via Monte Carlo samples and overlaid with the histogram of $u^*(Y,X,S)$. We show that the [[empirical histograms align well with the theoretical reference distribution on a simulation dataset with $\rho=0.8$]](https://anonymous.4open.science/r/ICLR2026-13252-DEFC/sim/check_mvnormal_2_0.8.pdf), and [[on the “Birth” dataset]](https://anonymous.4open.science/r/ICLR2026-13252-DEFC/real_data/checkbirths1.pdf). We will include a discussion of these diagnostics in the revised paper.
>
> 2.  We agree that evaluating FACTOR in higher-dimensional settings is important. We have conducted an additional experiment with $p = 6$ using the ”Air” data. The outcome vector includes:  max_PM2.5, max_NO2, max_O3, max_PM10, max_CO, max_SO2, and we continue to use Weekday as the protected subgroup variable. We compared FACTOR against MCP, HDR, and L-CP, and three other methods (ST-DQR, PCP, HD-CP) suggested by other reviewers. The [[results for $p=6$]](https://anonymous.4open.science/r/ICLR2026-13252-DEFC/real_data/real_data_example_p=6.pdf) show that: (i) FACTOR achieves the **second-smallest region size**, only slightly larger than HDR; (ii) FACTOR attains the **smallest KS distance** across all methods, indicating the best subgroup-calibration performance, and (iii) despite operating in six dimensions, FACTOR has the **second-shortest runtime**, second to L-CP, which does not enforce fairness constraints. We did not observe instability in optimization or calibration behavior relative to the $p=2$ cases. The [[full results for the real-world datasets]](https://anonymous.4open.science/r/ICLR2026-13252-DEFC/real_data/real_data.pdf), where the proposed method has similar performance on the "Wage" data with outcomes of $p=3$. We will include these results in the revision.

---

### Author Response · Authors · 2025-11-25
**Global Response**

Thank you to all four reviewers for your careful reading of our paper and constructive feedback. Before presenting point-by-point responses, we summarize the main strengths and concerns, and explain how our revisions and additional experiments address them.

**Strengths**:
1. [Reviewers `XxR7`, `wdLA`, `aMJP`, `vD5Z`] Method is novel, targets an important problem, and is well-motivated with applied examples
2. [`XxR7`, `wdLA`, `aMJP`] Carefully designed algorithm with creative implementation incorporating optimal transport and normalizing flows to optimize the KL divergence of the transformed distribution
3. [`XxR7`, `wdLA`, `vD5Z`] Nice theoretical guarantees: finite sample bound on subgroup calibration error that is $O(1/N_s)$, a significant improvement over marginal or asymptotic guarantees of most conformal methods
4. [`wdLA`, `aMJP`, `vD5Z`] Promising empirical results: target coverage, smaller prediction regions, and improved subgroup fairness than competitors in synthetic data and 6 real-world datasets
5. [`XxR7`, `aMJP`, `vD5Z`] Well-presented and clear methodology

**Concerns and our changes**
- [`XxR7`] The Complexity of implementation and robustness of the method to a breakdown in any component of the algorithm.
  - The ICNN used to learn the OT map has theoretical guarantees for universally approximating any convex function.
  - We have [[checked the quality of the learned OT map in simulated data]](https://anonymous.4open.science/r/ICLR2026-13252-DEFC/sim/check_mvnormal_2_0.8.pdf)  and [[real-world data]](https://anonymous.4open.science/r/ICLR2026-13252-DEFC/real_data/checkbirths1.pdf)  on the calibration sets.
- [`XxR7`, `vD5Z`] Increasing dimension of the outcome space beyond $p=2$.
  - We have conducted a [[new experiment with $p=6$ with the “Air” dataset]](https://anonymous.4open.science/r/ICLR2026-13252-DEFC/real_data/real_data_example_p=6.pdf), which shows very good training stability, competitive runtime, and improved prediction region size.
  - The "Wage" dataset now includes two continuous outcomes + one discrete outcome, and the [[full results are presented here]](https://anonymous.4open.science/r/ICLR2026-13252-DEFC/real_data/real_data.pdf).
- [`wdLA`, `aMJP`] Missing literature and comparison to other methods
  - We now discuss PCP by Wang et al (2023) and CONTRA by Fang et al (2025).  [[Additional methods are compared against in simulations]](https://anonymous.4open.science/r/ICLR2026-13252-DEFC/sim/sim_n5000.pdf) and [[real-world datasets]](https://anonymous.4open.science/r/ICLR2026-13252-DEFC/real_data/real_data.pdf) for comparison.
  - We now compare our work and our improvements in the related works, compared to Thurin et al (2025), Klein et al (2025), and high-density approaches, including Dheur et al (2024), Izbicki et al (2022), and Jonkers et al (2025).
- [`wdLA`, `aMJP`] Cut-off optimization directly in the outcome space with importance sampling
- [`aMJP`] Ablation study on the fairness constraints and the optimization of cut-off values
- [`aMJP`] More details about the actual creation of the prediction region with grid search
- [`wdLA`] Writing that makes it difficult to follow and check details
  - Related Work section is expanded to incorporate the suggested references (PCP, CONTRA, and high-density approaches), added missing contextual transitions, and rewrote several bullet-style segments into smoother prose with clearer explanations and notation.
  - We also added remarks throughout Sections 2 and 3 to guide readers and remove earlier ambiguities.

---

### Author Response · Authors · 2025-12-01
**Summary to AC**

Dear Area Chair,

We thank you for overseeing the review process. Below please find a summary of the significant progress we have made during the rebuttal period to assist in your decision-making, which we believe resolves the primary concerns raised by reviewers `wdLA` (Rating: 4) and `aMJP` (Rating: 4), while reinforcing the positive assessments of `XxR7` (Rating: 6) and `vD5Z` (Rating: 8). We have revised the paper and **uploaded the new PDF with changes highlighted in blue**, including new figures and comparator methods that reviewers have asked for.

1. Reviewer `XxR7` (Rating: 6) This reviewer appreciated the work and was quite positive while raising some minor concerns regarding **model misspecification** and **scalability**. In response, 1) we presented well-established theoretical guarantees for the model's goodness-of-fit and introduced a new empirical diagnostic to assess model fit on synthetic and real-world datasets; 2) Furthermore, we added more experiments to demonstrate the method's scalability and stability for higher outcome dimensions ($p=3$ and $6$), where FACTOR remained competitive in efficiency and fairness.

2. Reviewer `wdLA` (Rating: 4) This reviewer's concerns focused on **presentation clarity** and **the optimal cutoffs module**. 1) We have significantly improved the manuscript's clarity by refining transitions and aligning notation throughout. We also expanded the Related Work section to include other competitors and benchmarked against these new baselines in both synthetic and real-world settings; 2) Most importantly, we modified our cutoff selection module to directly minimize region size in the outcome space and updated the experiments throughout, addressing the reviewer's primary technical concern. We will also add that the other three reviewers gave the maximum score of 4/4 for "Presentation"; nevertheless, this reviewer's comments helped improve the presentation clarity further.

3. Reviewer `aMJP` (Rating: 4) This reviewer asked for **clearer differentiation from prior work** and **an analysis of the impact of fairness constraints**. 1) We expanded the Related Work section to better highlight FACTOR's unique guarantee of group-conditional coverage, a property confirmed by our theoretical and empirical comparisons; 2) In addition, we included a new ablation study demonstrating that enforcing fairness constraints by our method imposes minimal efficiency costs, which aligns with our theoretical expectations.

4. Reviewer `vD5Z` (Rating: 8) This reviewer strongly supported acceptance from the outset, praising the problem formulation and practicality. Our responses to their technical questions regarding **scalability** with $p> 2$, and **fairness measurements** further solidified their assessment, leading them to state, "Thanks for addressing my questions and concerns. I will be maintaining my score".

We believe these revisions have significantly improved the paper, making it well-positioned for the machine learning community. We hope this summary of our rebuttal will assist your assessment.

Best,
Authors

---

### Meta-Review · Area_Chair_xLny · 2026-01-07

**Summary:**

Major concern about the presentation: I agree with Reviewer wdLA that the writing style of the current version should be improved to make it more readable. The authors largely updated the manuscript (almost all the introduction and related work), but this paper would benefit from submission to another venue for re-evaluation on the improved manuscript. This ensures fair assessment of the revised version for this work.

**Reviewer Concerns:**

Addressed concerns:

1. Higher-dimensional scalability (p > 2): Addressed via new experiments on a p=6 dataset ("Air"), demonstrating competitive region size, KS distance, runtime, and training stability. Suggestions for low-rank ICNNs for even larger p were added.
2. Additional fairness metrics: Addressed by reporting the p%-rule metric (FACTOR closest to 1) and discussing extensions to other criteria like equalized odds.
3. Monotonicity assumption and cutoff optimization validity: Addressed by revising the method to directly minimize outcome-space volume via importance sampling with density weighting (updated Equations 5–6, Remark 3), eliminating reliance on monotonicity.

outstanding concerns: See summary.

**Reviewer Scores:**

Reviewers wdLA and aMJP may maintain their negative rates, while the other two maintain or increase their scores.

---

### Decision · Program_Chairs · 2026-01-26

Reject